

# Evaluating the Lower Tropospheric COSMIC GPS Radio Occultation Sounding Quality over the Arctic

Xiao Yu[1], Feiqin Xie*[1], Chi O. Ao[2]

[1]Department of Physical and Environmental Sciences, Texas A&M University – Corpus Christi, Corpus Christi, Texas,
78412, USA
[2]Jet Propulsion Laboratory, California Institute of Technology, Pasadena, California, 91011, USA

*Correspondence to: Feiqin Xie (Feiqin.Xie@tamucc.edu)

**Abstract.** Lower tropospheric moisture and temperature measurements are crucial for understanding weather predication and climate change. Global Positioning System radio occultation (GPS RO) has been demonstrated as a high-quality observation with high-vertical-resolution and sub-Kelvin temperature precision from the upper troposphere to the stratosphere. In the tropical lower troposphere, particularly the lowest 2 km, the quality of RO retrievals is known to be degraded and is a topic of active research. However, it is not clear whether similar problems exist in the high latitudes, particularly over the Artic, which is characterized by smooth ocean surface and often negligible moisture in the atmosphere. In this study, three-year (2008–2010) GPS RO soundings from COSMIC (Constellation Observing System for Meteorology, Ionosphere, and Climate) over the Arctic (65°N-90°N) show uniform spatial sampling with average penetration depth within 300 m above the ocean surface. Over 70% soundings penetrate below 300 m in all non-summer seasons but only about 50-60% in summer, when near-surface moisture and its variation increase. Both structural and parametric uncertainties of GPS RO soundings were also analyzed. The structural uncertainty (due to different data processing approaches) is estimated to be within 0.07% in refractivity, 0.72 K in temperature and -0.029 g/kg in specific humidity below 10 km, which is derived by comparing RO retrievals from two independent data processing centers. The parametric uncertainty (internal uncertainty of RO sounding) is quantified by comparing GPS RO with near-coincident radiosonde and ECMWF ERA-Interim profiles. A systematic negative bias up to ~1% in refractivity below 2 km is only seen in the summer, which confirms the moisture impact on GPS RO quality.

## 1 Introduction

Over the Arctic, the surface temperature has increased twice as much as global average rate in the past 100 years (Bernstein et al., 2007; Chae et al., 2015; Najafi et al., 2015), indicated by the decline of sea ice cover. The change of the





Arctic climate (e.g., temperature) along with the decline of sea ice cover are expected to affect the global climate (Vihma, 2014). Lower troposphere is one of most critical components for Arctic climate system, which has been intensely investigated by various observations (e.g., in-situ balloon sounding, ground-based, airborne, and satellite remote sensing). However, the lack of continuous high-vertical-resolution measurement in the Arctic lower troposphere impedes our

understanding of the complex physical processes that controls the air-sea-ice interaction, which is the key to improve Arctic weather forecasting and climate prediction.

Traditional radiosonde balloon soundings have long been the most reliable for sensing the atmospheric properties (e.g., temperature, pressure, and humidity) with high-vertical-resolution (Pelliccia et al., 2011) and widely used to calibrate and validate the satellite-borne retrievals (John and Buehler, 2005; Kuo et al., 2005). Over the Arctic, most of the radiosondes are

only sparsely available over the land near the Arctic-circle and few over the ocean from a handful of field campaigns. The ground-based and airborne remote sensing also suffer limitation in spatial and temporal coverage over the Arctic. The satellite remote sensing offers the opportunity for the uniform observation over the Arctic. The passive infrared sounders, such as AIRS (on Aqua), IASI (on MetOp A and B) and CrIS (on S-NPP), provides near daily and global vertical atmospheric profiles of temperature and moisture with ~1 to 2 km vertical resolution. However it cannot profile beneath the

clouds that cover the Arctic Ocean over up to 90%-95% in summer months and around 50% in winter (Zygmuntowska et al., 2012). The microwave sounders, such as the AMSU-A (on Aqua) and ATMS (on SNPP), could sense below the clouds, but also has coarse vertical resolution (~1-2 km) in the lower troposphere and encounter large uncertainty over land due to limited knowledge of the land emissivity (Deng et al., 2009; Weng et al., 2012).

Since the advent of the Global Positioning System (GPS) radio occultation (RO) technique in early 1990s, the RO

soundings have demonstrated a high-quality observation with sub-Kelvin temperature precision (Healy et al., 2005; Kursinski et al., 1997; Melbourne, 2004). After the launch of the six-satellite Constellation Observing System for Meteorology, Ionosphere, and Climate (COSMIC), GPS RO provides near real time, high-vertical-resolution, uniformly distributed global soundings of atmospheric bending angle and refractivity (Anthes et al., 2008) from stratosphere down to near the surface in all-weather condition. GPS RO bending angle and refractivity measurements have been operationally

assimilated into global weather forecasting models and demonstrate significant positive impact especially over the upper troposphere and above the open oceans (Cucurull et al., 2008; Healy et al., 2005). GPS RO measurements are also considered for global climate benchmark monitoring (Ho et al., 2009; Steiner, 2013) for their self-calibration and long-term stability (Steiner, 2013). Moreover, GPS RO has demonstrated its capability to observe the lower troposphere (Sokolovskiy et al., 2006a, 2010) and the planetary boundary layer (PBL) (von Engeln et al., 2005; Sokolovskiy et al., 2007; Ao et al.,

2012; Xie et al., 2012). The GPS RO measurement could fill the gap on observing the lower troposphere over the Arctic (e.g., Chang et al., 2017). Nevertheless, the uncertainty of RO sounding increases in the lower troposphere especially within the PBL, which remains largely uncharacterized over the remote Arctic Ocean.

One issue of GPS RO sounding for lower troposphere lies in that not all the RO profiles penetrate down to the surface (Ao et al., 2012; Xie et al., 2012). The limited penetration issue could be resulting from the early termination of RO



sounding due to the topographic blocking of RO signals grazing the Earth's surface. In addition, the corruption of RO signal due to the increasing receiver tracking error in the lower troposphere could also lead to early termination of the sounding profiles before reaching the surface. To measure the very shallow surface inversion that is often observed over the Arctic Ocean (Tjernström et al., 2014) requires very deep penetration of RO sounding. The implementation of open-loop tracking

technique, has significantly improved the fraction of soundings reaching within 1 km of Earth's surface (Ao et al., 2009). However, a significant fraction of the RO soundings still cannot reach the lowest 500 m above the surface, which could be especially problematic for sensing the Arctic shallow PBL, and warrants further studies.

Beyond the penetration issue, the structural uncertainty and parametric uncertainty affecting RO sounding quality should also be investigated. The structural uncertainty arises from different approaches of constructing the dataset from the same

raw data, whereas the parametric uncertainty is the uncertainty for the chosen approach in the presence of a finite sample of data (Thorne et al., 2005). Note that the NASA Jet Propulsion Laboratory (JPL) and the University Corporation for Atmospheric Research (UCAR) are two major independent COSMIC RO retrievals centers. The two centers use different inversion algorithms to derive the RO parameters from the same raw COSMIC RO measurements. The statistical comparisons between JPL and UCAR retrievals will shed some lights on the structural uncertainty of the COSMIC RO data.

Ho et al. (2009) and Steiner (2013) investigated the structure uncertainty of RO soundings from upper troposphere to the lower stratosphere (~8 to 25 km) among different data centers (including JPL, UCAR and several European centers), and revealed very small uncertainty (e.g., less than 0.06 K in temperature). In this study, the focus will be on assessing the structural uncertainty in the lower troposphere over the Arctic. In addition, the parametric uncertainty of COSMIC RO soundings will also be evaluated by comparing with independent radiosonde observations as well as the global reanalysis. A

better understanding of penetration issue along with the quantification of both structural and parametric uncertainties will help improve RO retrievals and further RO science applications.

This paper will focus on COSMIC RO soundings in the lower troposphere over the Arctic (65°N-90°N). Section 2 describes the COSMIC, radiosonde, and global reanalysis data used for this study. Section 3 details the definition of RO penetration depth and the RO retrieval difference between JPL and UCAR data centers. Section 4 presents seasonal variation

of penetration depth over the Arctic, and the RO structure and parametric uncertainties derived from the inter-center RO retrieval comparison between JPL and UCAR, and the comparison of RO retrievals with radiosonde and global reanalysis, respectively. Section 5 contains the summary and conclusions.

## 2    Data

This study analyzed Level-2 (e.g., refractivity, temperature and humidity) COSMIC RO data over the Arctic 65°N-90°N from two major RO data processing centers in the United States: JPL (http://genesis.jpl.nasa.gov/genesis/) and UCAR (www.cosmic.ucar.edu/cdaac/). In addition, radiosonde soundings from the Arctic Summer Cloud Ocean Study (ASCOS)





and the European Center for Medium-Range Weather Forecasts Reanalysis Interim (ERA-Interim, ERA-I) data over the Arctic are also used. The properties of the COSMIC RO, radiosonde, and ERA-Interim are listed in Table 1.

### 2.1    COSMIC GPS RO

Three years of COSMIC RO soundings from both JPL and UCAR over the Arctic 65°N-90°N were analyzed, with a special emphasis on the troposphere (e.g., below ~10 km). The post-processed version of the JPL retrievals and the UCAR retrievals from 2008 to 2010 were used, where a consistent retrieval algorithm has been implemented throughout the data processing period. The COSMIC RO data are grouped into four seasons, such as Winter-DJF (December-January-February), Spring-MAM, Summer-JJA, and Fall-SON. Note that month of December in the winter season is from previous year (e.g., DJF 2008 denoted as Dec. 2007, Jan. 2008, and Feb. 2008). The JPL retrieval algorithm has been detailed in Hajj et al. (2002) and Ao et al. (2012), whereas the UCAR retrieval algorithm can be found in Kuo et al. (2004). Different calibration (e.g., orbit and clock calibration, Schreiner et al., 2009; Wickert, 2002), retrieval algorithm, and quality control procedure (e.g., Ho et al., 2009) lead to some differences in the total throughput of COSMIC RO soundings from the two data centers. JPL retrievals generally yield slightly fewer soundings than UCAR retrievals. A total of 112,156 and 129,538 RO profiles are retrieved from JPL and UCAR, respectively. Additional discussions on the processing differences can also be found in Ho et al. (2009, 2012) and Steiner et al. (2013).

### 2.2    Radiosonde and global reanalysis

High-resolution radiosonde soundings from the ASCOS field campaign (red dots) throughout the cruise in the Atlantic sector of the Arctic from 3 August to 7 September 2008 (Tjernström et al., 2014), and the COSMIC RO (e.g., UCAR) soundings distribution (blue diamond) during the same time period over 75°N-90°N are showed in Fig. 1. Detailed description of ASCOS including the instrumentation and measurements can be found in Tjernström et al. (2014). Radiosondes were launched from the helipad of the icebreaker four times a day at approximately 00:00, 06:00, 12:00 and 18:00 Coordinated Universal Time (UTC). A total of 145 soundings were collected throughout the entire period of the cruise. Only one sounding was discarded due to the missing latitude and longitude information. Each profile was interpolated onto uniform vertical levels with intervals increasing from 5 m in the lowest 1 km of the troposphere to 100 m in the stratosphere (Birch et al., 2012). In addition, the ERA-I reanalysis profiles over the Arctic are also analyzed. The ERA-I applied T255 grid scheme with 0.75° horizontal grids (~83 km near equator or ~14 km near latitude ~80°N), and 60 vertical layers (Dee et al., 2011).



## 3    Methodology

In this section, COSMIC RO sounding penetration depth, the RO retrieval algorithm difference between JPL and UCAR centers, as well as the equal area gridding technique used for this study, will be discussed.

### 3.1    Penetration depth of COSMIC RO soundings

One major limitation of using GPS RO sounding for lower troposphere study is that not all the RO profiles reach the surface. Here we define the "penetration depth" as the minimum height of each individual RO sounding above the local

surface. Note that the GPS height is the height above the mean sea level (MSL), which is referred to the height above the geoid, i.e., the equipotential gravity surface height from a standard gravity model such as the Earth Gravity Model 1996 (EGM96). The high-resolution digital terrain elevation data (0.16° grid) used in this paper is also in reference above the geoid. Thereafter, the penetration depth in this paper will be the GPS height above local surface after subtracting the terrain elevation.

In the RO retrieval the penetration depth of an individual occultation sounding is affected by the quality of the GPS RO signal at the receiver and the quality control criteria used. Several factors could result in the degradation of RO signal that leads to earlier termination of RO profiles before the sounding reach the local surface, i.e., a positive penetration depth. For example, the topography along the GPS and the RO receiver line of sight could block the RO signals and lead to the early termination of the RO sounding. In addition, the RO signal could be degraded due to receiver signal tracking issues

attributed to the presence of large vertical moisture variation in the lower troposphere (Ao et al., 2012).

In the JPL retrieval system, the ending of an RO sounding (i.e., the penetration depth) is determined as the transition point where the RO signal quickly degraded into unusable noisy regime by fitting a step function to the transformed signal amplitude after the canonical transform inversion algorithm (Ao et al., 2012). A similar approach is applied in the UCAR retrieval as well but using a different transformed signal through Full spectrum Inversion (FSI) algorithm (Jensen et al.,

2003; Kuo et al., 2004). With the implementation of open-loop tracking in the COSMIC mission, the receiver tracking errors in the lower troposphere have been significantly reduced (Ao et al., 2003; Sokolovskiy et al., 2006a, 2006b). However, the non-uniform RO penetration depth across the globe remains and requires further investigation (Ao et al., 2012; Xie et al., 2012). It is important to note that the vertical resolution of the level-2 RO refractivity retrievals in the lower troposphere is limited to be ~200 m due to the vertical smoothing of the retrieval profiles. Therefore, the penetration depth at or below 100

m is essentially as good as reaching the surface.



### 3.2    COSMIC RO retrieval algorithm difference between JPL and UCAR data centers

The detailed description of the GPS RO technique has been covered extensively in several papers (Anthes et al., 2008; Hajj et al., 2002; Kursinski et al., 1997). Here we only summarize the key concepts of the retrieval processes. GPS RO
senses the atmosphere by tracking the GPS radio signals that traverse the atmosphere as a moving receiver sets (or rises) behind the horizon relative to the transmitting satellite. The radio wave is refracted and its travel time is delayed due to the variations of refractivity of the atmosphere. GPS RO precisely measures phase and amplitude of GPS signals that traverse the earth atmosphere. After the phases are calibrated by removing the GPS and LEO clock errors, a time series of excess phase at both GPS frequencies (e.g., L1 and L2) are derived. Then under the assumption of local spherically symmetric
atmosphere, vertical profile of the bending angle ($\alpha$) and the refractivity index ($n$) can be derived. In neutral atmosphere, the refractivity ($N=(n-1)$ x $10^6$) measured by GPS RO is related to pressure ($P$ in mbar), temperature ($T$ in K), and water vapor partial pressure ($Pw$ in mbar) as the following Eq. (1) (Smith and Weintraub, 1953):

$$N = a_1 \frac{P}{T} + b_1 \frac{P_w}{T^2} \qquad (1)$$

where constants $a_1$ = 77.6 K/mb, $b_1$ = 3.73 x$10^5$ K$^2$/mb.

Based on the RO refractivity, the temperature and humidity profiles can be derived with certain external information. In the upper troposphere and above, the second term (the so-called "wet term") on the right-hand-side of the refractivity Eq. (1) can be neglected. The so-called dry temperature can be derived such as follows:

$$N = a_1 \frac{P_{dry}}{T_{dry}} \qquad (2)$$

As the saturation vapor pressure decreases rapidly with decreasing temperature according to Clausius-Clapeyron equation,
the water vapor pressure $Pw$ can be neglected in the upper troposphere where temperature is low (e.g., $T$ < 250 K) (Hajj et al., 2002; Kursinski et al., 2000; Melbourne, 2004). Given the pressure (or more strictly, the dry pressure, $Pdry$, derived from the hydrostatic balance) and the RO refractivity profiles, the temperature can be retrieved. Accurate temperature profiles can be derived throughout the stratosphere down to mid-troposphere or even lower altitude depending on latitudes, where the water vapor is negligible.

In the middle or lower troposphere where the moisture is not negligible, the derivation of temperature and humidity becomes an underdetermined problem, which is generally referred as dry-wet ambiguity problem. To solve this issue, the JPL and UCAR data centers use different approaches.

In JPL retrieval (e.g., Hajj et al., 2002), the moisture retrieval only starts when the dry temperature is over 250 K, i.e., the moisture contribution to the refractivity becomes non-negligible (Kursinski et al., 1997). In the temperate and tropical
regions where the water vapor is abundant with large uncertainty, the temperature profile from Numerical Weather Prediction (NWP) model analysis usually is relatively better known, and thus can be used to aid the water vapor retrieval. In practice, the nearest 6-hourly ECMWF global analysis temperature and moisture profiles are interpolated into each RO





sounding location as a-priori. Given the RO refractivity profile ($Tdry$>250 K), the ECMWF temperature profile will then be used as a-priori to derive the RO water vapor profile. On the other hand, given the RO refractivity and the ECMWF water vapor, the RO temperature profile can also be derived. While the approach is relatively simple, the RO water vapor (or temperature) will contain both measurement uncertainty in RO refractivity and the uncertainty in the a-priori ECMWF
temperature (or water vapor).

UCAR data center applied the optimal estimation of the water vapor, temperature, and pressure through a variational method (Kuo et al., 2004). The variational method combines the occultation measurements (e.g., refractivity) with the a-priori (or background) atmospheric condition in a statistically optimal way (Healy and Eyre, 2000; Zou et al., 1995). For example, the optimal solution to the state vectors (e.g., $T$, $Pw$ and $P$) can be found by adjusting the state vector elements in a
way that is consistent with the estimated background errors, to produce simulated measurement values that fit the observations to within their expected observational errors (Healy and Eyre, 2000).

Both JPL and UCAR temperature and humidity retrievals require the a-priori information from models and thus are not independent measurement. Instead, measurements (e.g., refractivity, dry temperature) are model-independent observations. Note the errors of the geophysical parameters derived from the 1D-variational method (UCAR) could become more
challenging to interpret because the errors are combination of the a-priori model background errors with the RO measurement errors.

### 3.3    The equal-area-grid mapping method

Note that a fixed latitude-longitude grid (2.5° x 2.5°) near the equator has an area of ~78,400 km$^2$ that is equivalent to a square cell with side of ~280 km. However, the length of the fixed grid longitude is significantly reduced at higher latitude, especially near the polar region. To accommodate such a significant reduction of grid area at higher latitude resulting from the fixed latitude-longitude grid, the equal-area-grid mapping method is applied. A fixed 2.5° latitude interval (~280 km) is chosen, and each latitude band is evenly divided to have each grid area close to 78,400 km$^2$. Therefore, each grid will have a
roughly equal area across the polar region with a fixed 2.5° latitude interval and a variable longitude interval (increasing at higher latitude or toward the poles). The mapping technique will result in only 3 grids within 87.5°N-90°N latitude bands, 9 grids within 85°N-87.5°N and increasing further south to be the maximum of 58 grids within 65°N-67.5°N (Rossow and Schiffer, 1991).

### 4    Results and discussions





In this section, spatial and temporal variation of the COSMIC RO sounding penetration depth, the structural uncertainty of RO retrieval difference between JPL and UCAR centers, the parametric uncertainty between RO retrievals and the radiosonde as well as ERA-I reanalysis will be discussed.

4.1     Spatial and temporal variation of RO penetration depth

COSMIC RO refractivity profiles over the Arctic (65°N-90°N) from both JPL and UCAR are analyzed. The three-year (2008-2010) median seasonal variations of the penetration depth for both centers and their associated RO profile sample numbers are shown in Fig. 2. Both centers show rather homogeneous sampling with ~60-120 soundings in each equal-area
grid (equivalent to the area of a square cell of 280 km x 280 km). JPL shows slightly fewer sounding at each grid than UCAR, which is consistent with the overall smaller sample size.

Very similar geographical pattern of RO penetration depth is seen for both centers in all seasons, with rather uniform deep penetration (~0-300 m) over the Arctic Ocean, and generally poorer penetration over land and islands. The distinctly poorer penetration (i.e., larger penetration depth) in summer than other seasons, indicates the impact of RO retrieval from the
increase of lower tropospheric water vapor over the warmer ocean in summer. In addition, central Greenland shows deep penetration (100-300 m) likely due to the relatively flat terrain. Over the land surrounding the Arctic Ocean, the penetration depth varies from 100-900 m with slightly poorer penetration in JPL data, over the north-western edge of the Greenland. It is worth noting that ~100m penetration depth is essentially as good as reaching the surface as both JPL and UCAR centers apply ~200 m vertical smoothing on the RO bending/refractivity retrieval.

Figure 3 further illustrates the seasonal-mean percentage of RO profiles reaching different altitude (binned into 100-m vertical intervals) above the surface with land and ocean separated. For both centers, high percentage of COSMIC RO soundings (~85-90%) penetrates below 1 km in all seasons. But a sharp decrease by ~8-20% is seen in RO profiles reaching 0.5 km above the surface. Above 1 km, no significant difference in RO penetration is seen between the land and the ocean. However, a slight decrease in the percentage of RO profiles reaching below 1 km over land could be due to topography
blocking effect.

The UCAR retrieval shows generally higher percentage profiles (~92-95%) penetrating down to 1 km above the surface than the JPL (~88%) in all seasons except in the summer when the percentage drops to ~84% for both centers. UCAR also shows generally higher percentage RO soundings extending to the height levels within 0.5 to 1 km except the summer season when JPL shows slightly higher percentage of profiles extending to all levels below 1 km. In addition, JPL also shows
slightly higher percentage RO profiles extending below 0.3 km than UCAR in all seasons with a maximum of ~5% difference in summer.





The much higher percentage of deep penetrating RO profiles below 0.5 km over the very dry Arctic Ocean (65-80%) than the moist tropics of 20-30% (Ao et al., 2012; Xie et al., 2012) strongly indicates that the increase of lower tropospheric moisture could result in larger receiver tracking errors and the early termination of the RO profile. The poorer penetration in the warmer and moister summer Arctic further confirms the moisture impact on the RO penetration issue.

### 4.2    Inter-center comparisons between JPL and UCAR retrievals

To quantify the structural uncertainty of the COSMIC RO soundings over the Arctic (65°N-90°N), the RO retrievals difference between JPL and UCAR data centers are analyzed. The RO soundings in both winter (DJF) and summer (JJA)

seasons of 2008 are investigated. Although both data centers start from the same level 1 raw COSMIC measurements, their retrieval algorithms are slightly different in terms of calibration, retrieval, and quality control processes (Ho et al., 2012). A total of 4782 pairs of common soundings in winter (DJF) and 8375 pairs in summer (JJA) of 2008 have been identified.

All RO sounding refractivity, temperature and specific humidity (e.g., $N$, $T$, $q$) from both centers are interpolated into 100 m vertical intervals. The ensemble mean, standard deviation, standard error of the inter-center differences and the

sampling number are shown in Fig. 4. The statistics at three selected altitude ranges (e.g., from surface up to 3 km, 5 km and 10 km, respectively) are summarized in Table 2. It is also worth to note that the significantly decreasing number of common refractivity soundings in the lowest 3 km, is mainly due to the limited RO sounding penetration depth as discussed in section 4.1. The statistics for the lowest 200 m were discarded due to the significant drop of the number of RO observations near the surface, limited by the ~200 m vertical resolution of RO retrievals. For the RO temperature and specific humidity profiles,

the variation of common sounding numbers at different altitudes is mainly limited by the availability of JPL retrievals. For example, UCAR retrieves both $T$ and $q$ at all altitude where refractivity retrievals are available. However, JPL only retrieves tropospheric temperature at altitudes cooler than a threshold of about 250 K (see Sec. 3.2), leading to significant reduction in the number of JPL temperature retrievals in the lower troposphere, especially in the warmer summer season (Fig. 4d). To account for the number of pairs change with the altitude, the standard error is also computed (i.e., the standard deviation

divided by the square root of the sample size), which is an estimation of the difference of the sample mean from the population mean affected by the sample size.

In winter, small but persistent biases are seen in JPL retrieval as compared to the UCAR retrieval below 3 km in refractivity (~0.04%), temperature (0.54 K) and specific humidity (0.03 g/kg), with their standard deviation of 0.19%, 0.45 K, 0.33 g/kg, respectively. In summer, the difference for all three parameters slightly changes with the mean difference of

0.02%, 0.59 K, -0.05 g/kg, and the corresponding standard deviation of 0.44%, 0.37 K, 0.50 g/kg, respectively.

The mean fractional refractivity difference between JPL and UCAR is ~0.06% in winter and 0.07% in summer below 10 km, which is consistent in magnitude with the inter-center comparison of CHAMP RO data within 8-12 km over 60°N-90°N (Ho et al., 2009). No significant difference is seen between the winter and the summer. The mean temperature difference is





0.38 K in winter and 0.61 K in summer below 10 km, with the corresponding standard deviation of 0.40 K and 0.62 K, respectively. The mean specific humidity difference is 0.01 g/kg in winter and near zero in summer below 10 km with the corresponding standard deviation of 0.31 g/kg and 0.34 g/kg, respectively. The standard errors are generally small, except when there is a sharp decrease in sample size (e.g., Fig. 4d and 4e). The generally small standard error indicates reliable
statistics given the large sample size at most altitude levels.

### 4.3    Comparisons of the radiosonde with the RO retrievals and the ERA-I profiles

To quantify the parametric uncertainty of the COSMIC RO soundings over the Arctic (65°N-90°N), both JPL and
UCAR  retrievals are collocated with a total of 144 radiosonde soundings collected from ASCOS summer field campaign in 2008. A total of 65 JPL profiles and 68 UCAR profiles were found to be collocated with the radiosondes within 3 hours and 300 km. The ensemble mean difference, standard deviation, and standard errors are shown in Fig. 5. The statistics of the comparisons at three selected altitude range (e.g., 3 km, 5 km and 10 km from the surface) are listed in Table 3.

Comparing with the radiosonde from surface up to 10 km, the COSMIC RO refractivity show an overall median bias of
0.10% in JPL retrievals (Fig. 5a$_1$), -0.20% median bias in UCAR retrievals (Fig. 5a$_2$), and -0.21% overall median bias in ERA-I refractivity retrievals (Fig. 5a$_3$). However, the overall JPL median bias (0.10%) is comprised of both positive bias (e.g., ~0.2% in 6-8 km) and negative bias (e.g., ~ -0.5% below 4 km). Most of the negative bias is coming from the lower troposphere in JPL and UCAR. For example, JPL refractivity retrievals show a negative bias below ~2 km, which increases to a maximum of about -1% in the lowest 1 km above the surface. Similarly, UCAR refractivity retrievals also show a
negative bias but starting from ~5 km down to the surface with a smaller maximum bias of about -0.5%. On the other hand, the ERA-I shows generally negative above ~1 km but a small positive mean bias below ~0.5 km.

Figure 5b$_1$ shows a warm bias of 0.62 K in JPL temperature retrievals from 10 km down to ~4.5 km where JPL stop retrieving temperature. A sharp increase in standard error in JPL temperature below 5 km are due to the sharp drop in the available JPL temperature retrieval. On the other hand, UCAR temperature shows a smaller overall median bias of 0.53 K
below 10 km, with an increasing bias below 2 km up to near 2 K (Fig. 5b$_2$). ERA-I temperature shows a smallest overall median bias of 0.12 K below 10 km (Fig. 5b$_3$) with a warm bias (~1K) below 1 km and small cold bias above, which are consistent with Wesslén et al. (2014).

For specific humidity, the JPL retrievals exhibit negative biases from surface up to ~2 km and transition to positive biases above, with a median bias of about 0.015 g/kg from surface up to ~6.5 km (Fig. 5c$_1$). The UCAR specific humidity
has a negative overall bias of -0.028 g/kg and mostly coming from below ~5 km (Fig. 5c$_2$). The ERA-I specific humidity has an overall bias of 0.004 g/kg and mostly coming from below ~0.6 km (Fig. 5c$_3$). It is also worth noting that the increasing biases and variations of RO refractivity, temperature and specific humidity retrievals below 5 km, strongly indicate the impact of the increasing availability and variation of water vapor in the Arctic summer on the RO retrievals.





Overall, JPL and UCAR refractivity retrievals are consistent with temperature and humidity retrievals (e.g., negative refractivity bias corresponding to positive bias of temperature and negative bias of the specific humidity), except at certain altitudes, JPL and radiosonde comparison showing positive fractional refractivity bias corresponding to positive temperature bias (e.g., 6-8 km) that may be affected by pressure. In JPL retrieval, the refractivity errors are directly mapped into both

temperature and humidity errors. In UCAR 1D-variational retrieval, the refractivity errors can be mapped to either or both temperature and humidity errors. In addition, the errors in the a-priori information from ECMWF model analysis used for both JPL and UCAR will also affects the RO temperature and humidity retrieval. It is worth noting that the errors of the geophysical parameters derived from the 1D-variational method in UCAR retrieval become more challenging to interpret as they includes errors from both model background and the RO measurement. For instance, the errors of UCAR specific

humidity are very similar to the ERA-I but not consistent with the refractivity errors, which indicate the model a-priori humidity might dominate the 1D-variational UCAR humidity retrieval.

It is important to point out that the sharp drop in the number of near-coincidence profiles below ~1 km (Fig. 5) is primarily due to the limited number of RO soundings penetrating deep into the PBL, especially the bottom ~300 m above the surface.

## 4.4    Comparisons between the RO retrievals and the ERA-I profiles

To further quantify the parametric uncertainty, the COSMIC RO soundings from JPL and UCAR are also compared with the near-coincident ERA-I reanalysis profiles. As the ERA-I assimilated COSMIC RO bending angles retrieved by UCAR data center, they are not fully independent datasets. However, in the data assimilation, large RO measurement errors

at lower altitudes (e.g., 20% in bending angles errors near the surface) are normally applied, along with the limited number of available RO soundings, the impact of RO sounding on ERA-I in the lower troposphere remains limited (Poli et al., 2010).

The total numbers of common COSMIC RO soundings from both JPL and UCAR are 4782 pairs in winter and 8375 pairs in summer of 2008 over the Arctic (65°N-90°N), respectively. The comparison between COSMIC RO from the two centers and their near-coincident ERA-I profiles are presented in terms of fractional refractivity (Fig. 6), temperature (Fig. 7)

and specific humidity (Fig. 8) differences. All profiles are interpolated into 100 m vertical intervals before the difference statistical calculation. Again, the statistics for the lowest 200m were discarded due to the significant drop of the number of RO observations near the surface. The statistical differences between COSMIC RO and ERA-I at three selected altitude ranges (e.g., 3 km, 5 km and 10 km from the surface) are detailed in Table 4.

In Figure 6, the fractional refractivity difference between RO (JPL and UCAR) profiles and the near-coincident ERA-I

profiles show small biases in winter. In the lowest 3 km, JPL refractivity exhibits a small positive bias of ~0.12%, whereas UCAR has a smaller positive bias of ~0.07%. In summer, an increase in positive bias above ~4 km and negative bias below ~2 km are seen. The maximum negative bias reaches about -0.7% (-0.6%) in JPL (UCAR) retrieval near 0.5 km. An averaged negative refractivity bias of about -0.3% below 3 km is seen in both centers.


Similarly, Figure 7 shows the temperature difference between COSMIC RO and the near-coincident ERA-I profiles. Below 3 km, in winter, JPL shows a positive bias of 0.14 K, whereas UCAR has a negative bias of -0.12 K. In summer, UCAR has a warm bias of 0.58 K, whereas JPL do not have retrieval below 3 km. Below 10 km, temperature difference in winter (Fig. 7a,c) exhibits much smaller variation comparing with that in summer. For instance, relative large positive biases

are seen in temperature retrievals from UCAR below 2 km (Fig. 7d) and from JPL within 6-10 km (Fig. 7b) comparing with the ERA-I profiles.

Moreover, Figure 8 shows the difference between the specific humidity retrievals from COSMIC RO and the near-coincident ERA-I. Within 3 km, JPL and UCAR have a negligible bias of -0.002 g/kg and -0.019 g/kg, respectively, in winter. In summer, much larger dry biases of about -0.124 g/kg in JPL and -0.083 g/kg in UCAR retrievals are detected.

Overall, the RO specific humidity exhibits negative biases below 2 km, likely due to the abundant water vapor near the ocean surface.

Overall, the COSMIC RO (JPL or UCAR) refractivity difference from the near-coincident ERA-I retrievals are consistent with the temperature and humidity retrievals differences (e.g., negative refractivity bias corresponding to positive bias in temperature and negative bias in the specific humidity). On the contrary, positive fractional refractivity bias

corresponds to positive temperature bias in JPL retrievals. Most of mean biases have negligible standard errors due to large sample sizes.

4.5     Case study of RO signal dynamics

To further study the impact of moisture on the RO soundings, the L1 SNR and excess Doppler for two typical JPL COSMIC RO soundings from summer (a, b) and one from winter (c) were presented along with the near-coincident ASCOS radiosonde (in summer) and the ERA-I profiles (in winter) (Fig. 9). The two summer cases shows near doubled moisture with specific humidity (~2 g/kg) near surface than that in the winter ( less than ~1 g/kg). In the winter case, very smooth excess Doppler along with a relatively quiet SNR are shown. A sharp drop in SNR to the noise background is clearly seen

around 39 seconds, when tangent point descends to the smooth Arctic ocean surface. On the contrary, much larger variations in both the SNR and excess Doppler are seen in the two summer cases especially below 5 km. The transition of the SNR to noise background near surface is smeared due to more lower troposphere moisture variations, which likely introduce multipath and SNR variations. Even though the lower troposphere moisture in Arctic summer is still rather low (less than 2 g/kg near surface) as compared to the low latitudes, a surprisingly large difference in refractivity (−7% near surface) is seen



(Fig. 9a2). The systematic negative RO *N*-bias (−1%) in summer season (e.g., Figs. 5&6) could be directly attributed to the lower troposphere moisture variations. However, the impact of moisture and its variations on the RO signal dynamics and so the RO calibration and retrieval processes warrant a more comprehensive investigation.

## 5    Summary and conclusions

In summary, over the Arctic (65°N-90°N), three-year (2008-2010) COSMIC RO soundings show uniform spatial sampling with average penetration depth (the minimum profile height) within 300 m above the ocean surface. Over 70% of COSMIC soundings penetrate deep (within 300 m) in all non-summer seasons but only 50-60% in the summer. The increase

of the near-surface moisture and its variation in summer, even though relatively small comparing to the tropics, can lead to significant GPS RO SNR and excess Doppler variations, which could complicate the GPS RO signal tracking and lead to early sounding termination before reaching the surface.

Both structural uncertainty and parametric uncertainty of COSMIC RO soundings have been quantified. The structural uncertainty of RO is estimated by comparing the retrieved refractivity, temperature and specific humidity from JPL and

UCAR processing centers, which process the same raw COSMIC GPS data. The comparisons using one-year COSIMC data in 2008 show the inter-center RO retrieval difference (i.e., structural uncertainty) within ~0.07% in refractivity, ~0.72 K in temperature and ~0.029 g/kg in specific humidity below 10 km. The parametric uncertainty is quantified by comparing RO with the near-coincident radiosonde and the ERA-I reanalysis. COSMIC RO shows slightly larger difference from the near-coincident radiosondes than the ERA-I, which assimilated UCAR COSMIC RO retrievals. A systematic negative bias up to

~1% in refractivity below 2 km is only observed during the summer, which further confirms the impact of the lower tropospheric summer moisture on RO retrievals. The parametric uncertainty of the COMIC RO refractivity sounding in summer season is about 2 orders of magnitude larger than the structural uncertainty, implying the highly consistent, precise COSMIC RO observations in the troposphere. It is reasonable to expect the parametric uncertainty in the winter season should be even smaller due to much less impact of moisture on the RO retrievals.

In conclusion, GPS RO provides high quality measurement (especially in refractivity) in the lower troposphere over the Arctic. The high-precision RO measurements with uniform spatial and temporal sampling provide promising opportunity for studying the lower tropospheric dynamic process, especially the PBL study. However, the early termination of RO sounding before reaching the surface and the systematic RO refractivity bias inside the PBL in summer limit the RO sounding capability inside the PBL, and impede its application for the physical process study involving the interaction of ocean,

atmosphere and sea ice. Preliminary study shows the impact of moisture on the RO signal dynamics. Further study is needed to improve the RO sounding quality and enhance the scientific application of RO observations in the lower troposphere.





**Acknowledgments**

This research was partially supported by NASA grant NNX14AK17G, NSF grant AGS-1262041 and the Research Commercialization Outreach Office of Texas A&M University-Corpus Christi. The authors also acknowledges the NASA Jet Propulsion Laboratory (JPL) Education Office for the summer internship opportunity in supporting the lead author.

5   COSMIC GPS radio occultation soundings were obtained from the Jet Propulsion Laboratory (https://genesis.jpl.nasa.gov/genesis/), and the University Corporation for Atmospheric Research (http://cdaac-www.cosmic.ucar.edu/cdaac/products.html). The ASCOS radiosonde soundings were provided by Professor Michael Tjernström from Stockholm University. The ERA-Interim data were obtained from ECMWF (http://apps.ecmwf.int/datasets/data/interim-full-daily/levtype=sfc/).



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





**Table 1. The properties of the COSMIC RO, radiosonde, and ERA-Interim.**

| Data Type | | Date | Parameters | Region | Vertical Resolution | Horizontal Resolution |
|---|---|---|---|---|---|---|
| COSMIC RO | UCAR | 2007-2010 | Refractivity, temperature & humidity | 65°N-90°N | ~200 m | ~200 km |
| | JPL | 2007-2010 | | | | |
| Radiosonde | ASCOS[a] | 03/08-07/09 2008 | Pressure, temperature & humidity | ~78°N-87°N | ~5-100 m | |
| ERA-Interim[b] | | 2007-2008 | Pressure, temperature & humidity | 65°N-90°N | 60-level (28-level below ~10km) | 0.75°latitude ×0.75°longitude |

[a] ASCOS: Arctic Summer Cloud Ocean Study field campaign

[b] ERA-Interim: European Center for Medium-Range Weather Forecasts Reanalysis Interim



**Table 2. Statistical comparison between JPL and UCAR retrievals from COSMIC RO**

| Seasonal Mean Difference | Altitude Range (km) | DJF 2008 (4782 pairs) $\mu\,(\sigma)$ * | JJA 2008 (8375 pairs) $\mu\,(\sigma)$ |
|---|---|---|---|
| $(N_{JPL} - N_{UCAR})/N_{UCAR}$ (%) | 0-3 | 0.041(0.193) | 0.016(0.438) |
| | 0-5 | 0.049(0.181) | 0.043(0.383) |
| | 0-10 | 0.062(0.189) | 0.072(0.328) |
| $T_{JPL} - T_{UCAR}$ (K) | 0-3 | 0.537(0.452) | 0.591(0.369) |
| | 0-5 | 0.506(0.465) | 0.723(0.597) |
| | 0-10 | 0.376(0.402) | 0.605(0.620) |
| $q_{JPL} - q_{UCAR}$ (g/kg) | 0-3 | 0.029(0.332) | -0.051(0.502) |
| | 0-5 | 0.027(0.344) | -0.002(0.414) |
| | 0-10 | 0.010(0.312) | -0.000(0.340) |

*Note that $\mu$ and $\sigma$ represent mean and standard deviation, respectively.





**Table 3. Statistical comparisons between COSMIC RO (JPL/UCAR) and the near-coincident Radiosonde .**

| Mean Errors | Altitude Range (km) | JPL-RDS (65 pairs) | UCAR-RDS (68 pairs) | ERAI-RDS (144 pairs) |
|---|---|---|---|---|
| | | $\mu$ $(\sigma)$ | $\mu$ $(\sigma)$ | $\mu$ $(\sigma)$ |
| $\Delta N/N_{RDS}$ (%) | 0-3 | -0.428(1.733) | -0.496(1.580) | -0.117(1.005) |
| | 0-5 | -0.285(1.458) | -0.360(1.337) | -0.212(0.939) |
| | 0-10 | -0.097(1.121) | -0.201(0.991) | -0.205(0.673) |
| $\Delta T$ (K) | 0-3 | | 1.047(2.318) | -0.170(1.216) |
| | 0-5 | | 0.741(2.149) | -0.051(1.065) |
| | 0-10 | 0.619(1.940) | 0.532(2.055) | 0.117(0.956) |
| $\Delta q$ (g/kg) | 0-3 | -0.158(0.667) | -0.081(0.593) | -0.003(0.388) |
| | 0-5 | -0.068(0.547) | -0.075(0.492) | -0.016(0.338) |
| | 0-10 | 0.015(0.480) | -0.028(0.289) | 0.004(0.199) |





**Table 4. Statistical comparisons between COSMIC RO (JPL/UCAR) and the near-coincident ERA-Interim.**

| Mean Difference | Altitude Range (km) | UCAR-ERAI DJF (4782 pairs) | UCAR-ERAI JJA (8375 pairs) | JPL-ERAI DJF (4782 pairs) | JPL-ERAI JJA (8375 pairs) |
|---|---|---|---|---|---|
| | | $\mu\ (\sigma)$ | $\mu\ (\sigma)$ | $\mu\ (\sigma)$ | $\mu\ (\sigma)$ |
| $\Delta N/N$ (%) | 0-3 | 0.070(0.784) | -0.259(1.634) | 0.121(0.784) | -0.253(1.634) |
| | 0-5 | 0.078(0.696) | -0.107(1.449) | 0.132(0.696) | -0.071(1.449) |
| | 0-10 | 0.068(0.616) | -0.054(1.038) | 0.133(0.616) | 0.123(1.038) |
| $\Delta T$ (K) | 0-3 | -0.120(1.829) | 0.580(2.254) | 0.143(1.829) | |
| | 0-5 | -0.132(1.629) | 0.313(2.001) | 0.196(1.629) | |
| | 0-10 | -0.135(1.358) | 0.109(1.649) | 0.153(1.358) | 0.246(1.649) |
| $\Delta q$ (g/kg) | 0-3 | -0.019(0.171) | -0.083(0.628) | -0.002(0.171) | -0.124(0.628) |
| | 0-5 | -0.013(0.134) | -0.056(0.507) | -0.001(0.134) | -0.053(0.507) |
| | 0-10 | -0.007(0.075) | -0.028(0.291) | -0.001(0.075) | -0.024(0.291) |



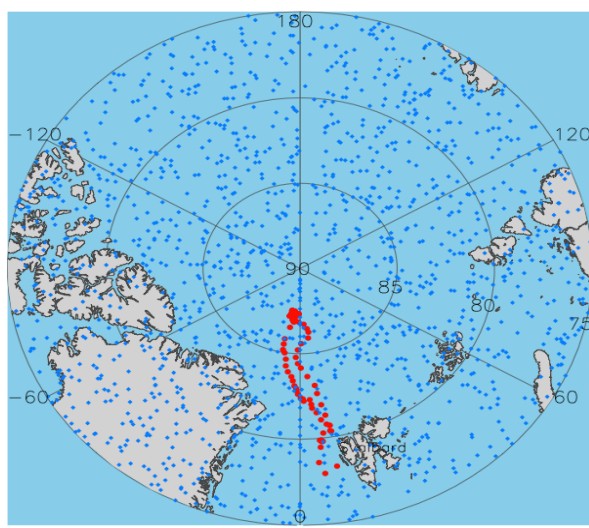

**Figure 1: Radiosonde soundings track (red dot) during ASCOS field campaign from 3 August to 7 September 2008, and the COSMIC RO (e.g., UCAR) soundings distribution (blue diamond) during the same time period over 75°N-90°N.**





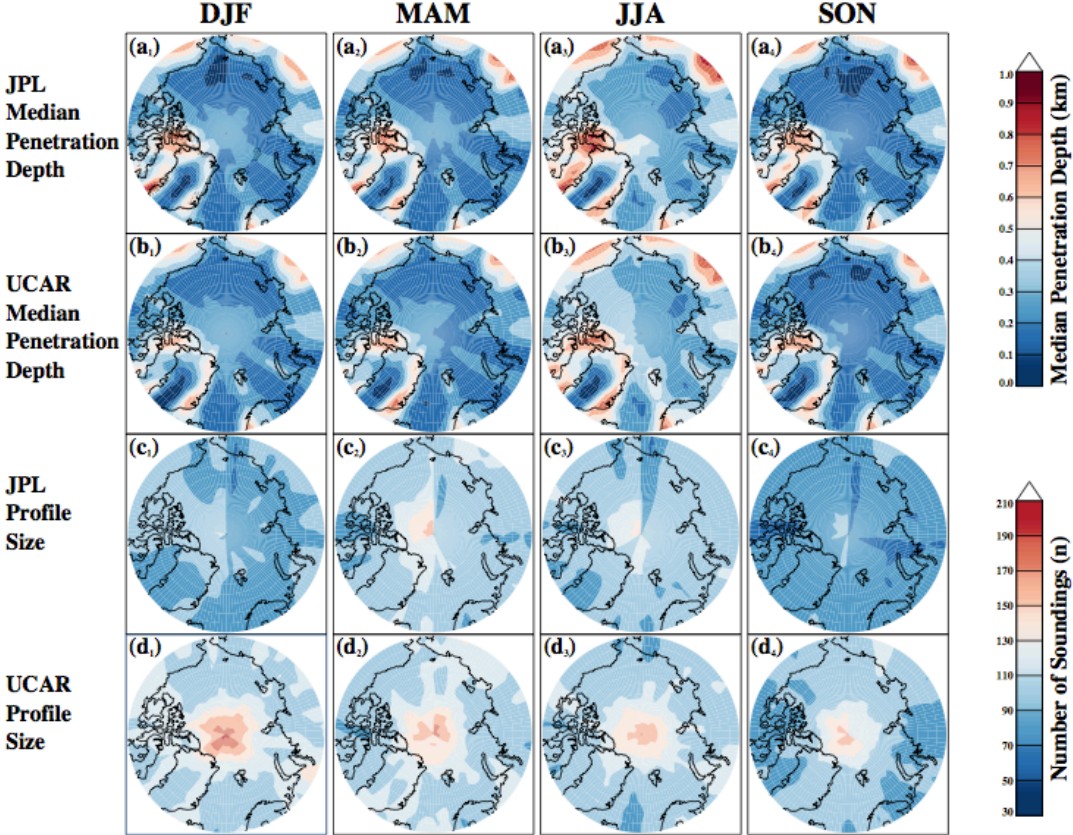

Figure 2: Seasonal median penetration depth of COSMIC RO soundings for JPL ($a_1$-$a_4$) and UCAR ($b_1$-$b_4$) and the corresponding number of soundings in each season for JPL ($c_1$-$c_4$) and UCAR ($d_1$-$d_4$) over the Arctic (65°N-90°N) from 2008 to 2010. Note the equal area mapping uses the equivalent square cell with side of ~280 km.





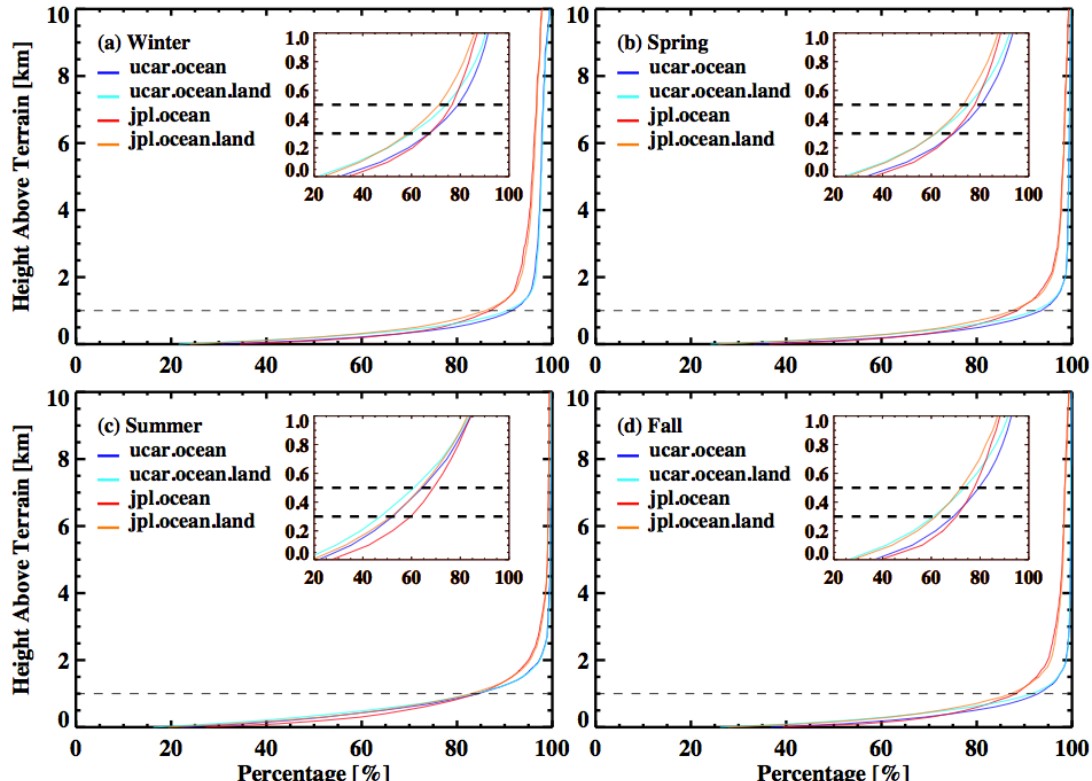

Figure 3: Seasonal-mean percentage of COSMIC RO profiles penetrating into the troposphere above the terrain over the Arctic
65°N-90°N. Darker and lighter blue describe UCAR profiles over ocean only and over both ocean and land, respectively. Red and
5    orange describe JPL profiles over ocean only and over both ocean and land, respectively. The inlet plots show the lowest 1 km with
the two dashed lines marking the heights of 0.3 km and 0.5 km.



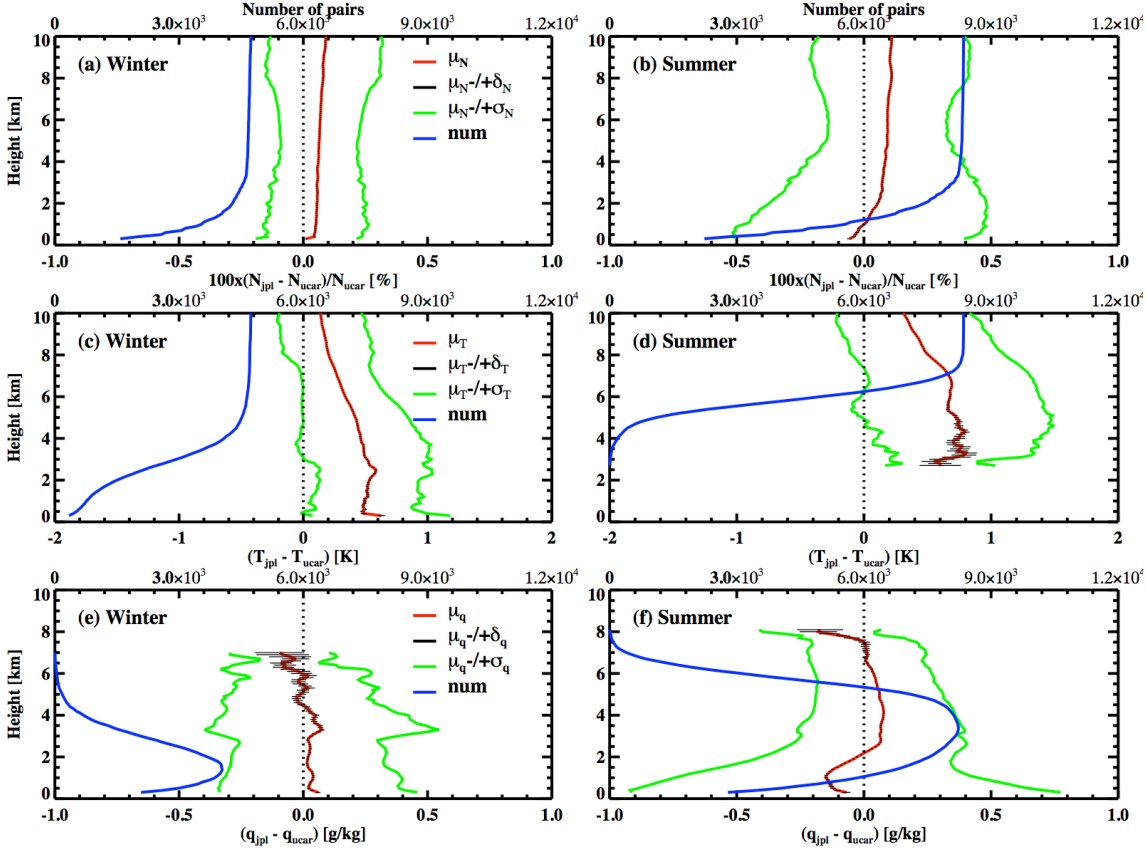

**Figure 4:** Statistical comparisons of COSMIC RO between JPL and UCAR retrievals are showed in fractional refractivity (a, b), temperature (c, d) and humidity (e, f) over the Arctic (65°N-90°N) in winter-DJF (a, c, e) and summer-JJA (b, d, f) of 2008. The mean difference ($\mu$, red), the mean plus-minus ($\pm$) one standard deviation ($\sigma$, green) and standard error ($\delta$, black horizontal bar) for all three parameters are also shown. The number of common sounding pairs varies with heights is shown (blue) with scale marked on the top to each panel.



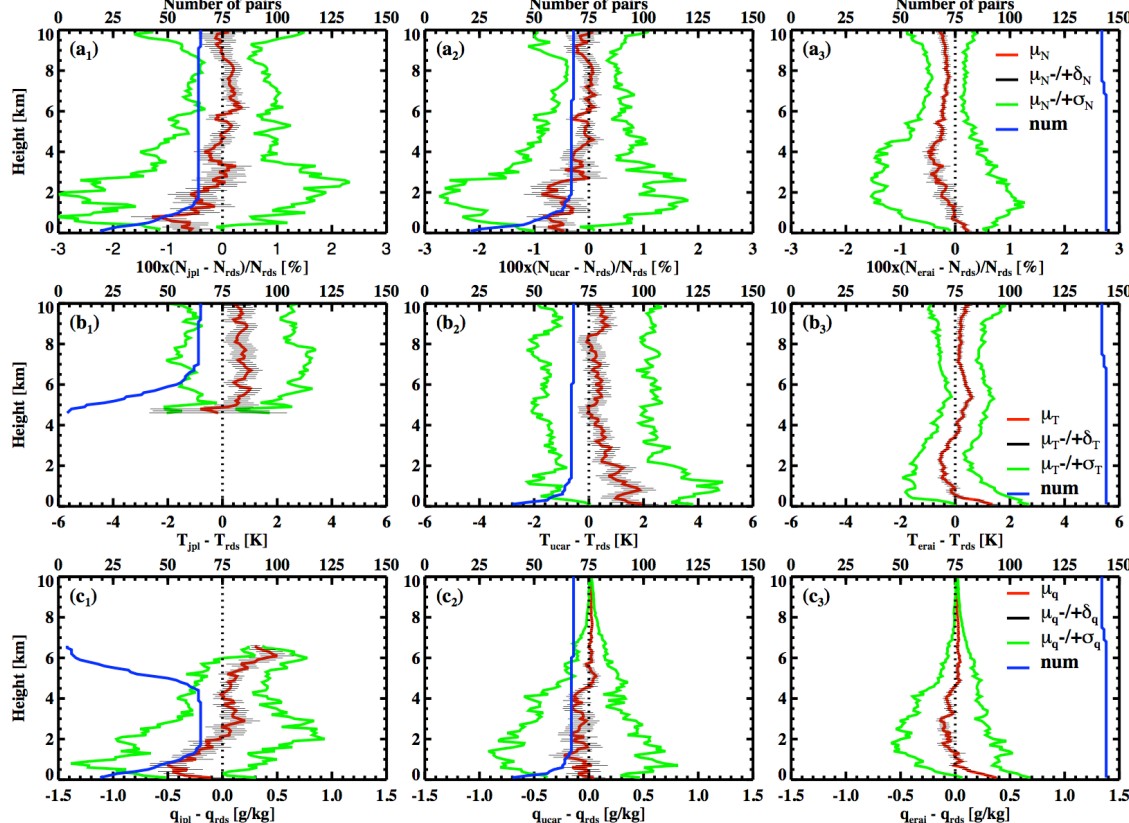

**Figure 5: Difference between ASCOS radiosonde soundings and the near-coincident COSMIC RO from JPL (left), UCAR (middle) and ERA-I (right) in terms of fractional refractivity ($a_1$-$a_3$), temperature ($b_1$-$b_3$) and specific humidity ($c_1$-$c_3$). The median difference ($\mu$, red), the median difference plus-minus ($\pm$) median absolute deviation ($\sigma$, green) and median-standard-error ($\delta$, gray horizontal bar) for all three parameters are also shown. The number of near-coincident pairs as a function of height is shown in blue with scale marked on the top to each panel.**

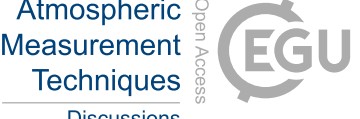



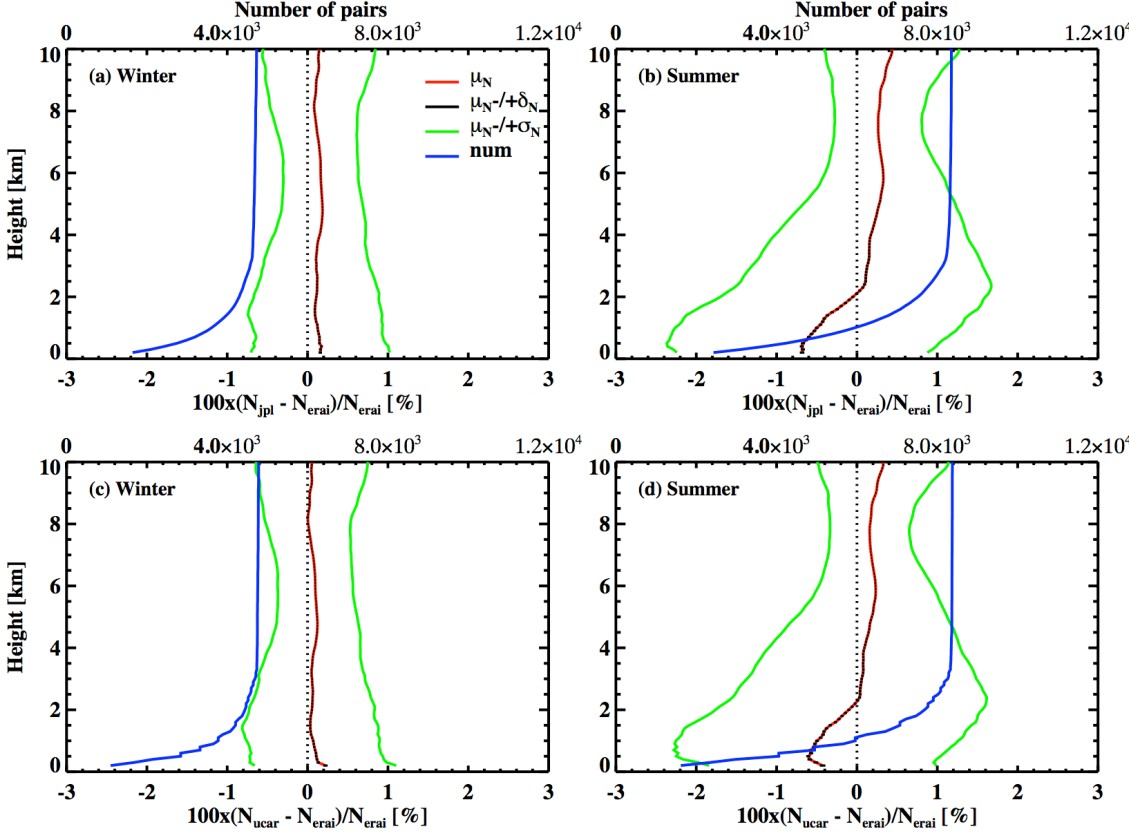

**Figure 6: Fractional refractivity difference between COSMIC RO (JPL–top/UCAR–bottom) and the near-coincident ERA-I reanalysis over the Arctic (65°N-90°N) during winter (DJF, left) and summer (JJA, right) of 2008. The mean difference (red), mean plus/minus one standard deviation (black), and the number of RO profiles that penetrated down to a given altitude (blue) are shown.**




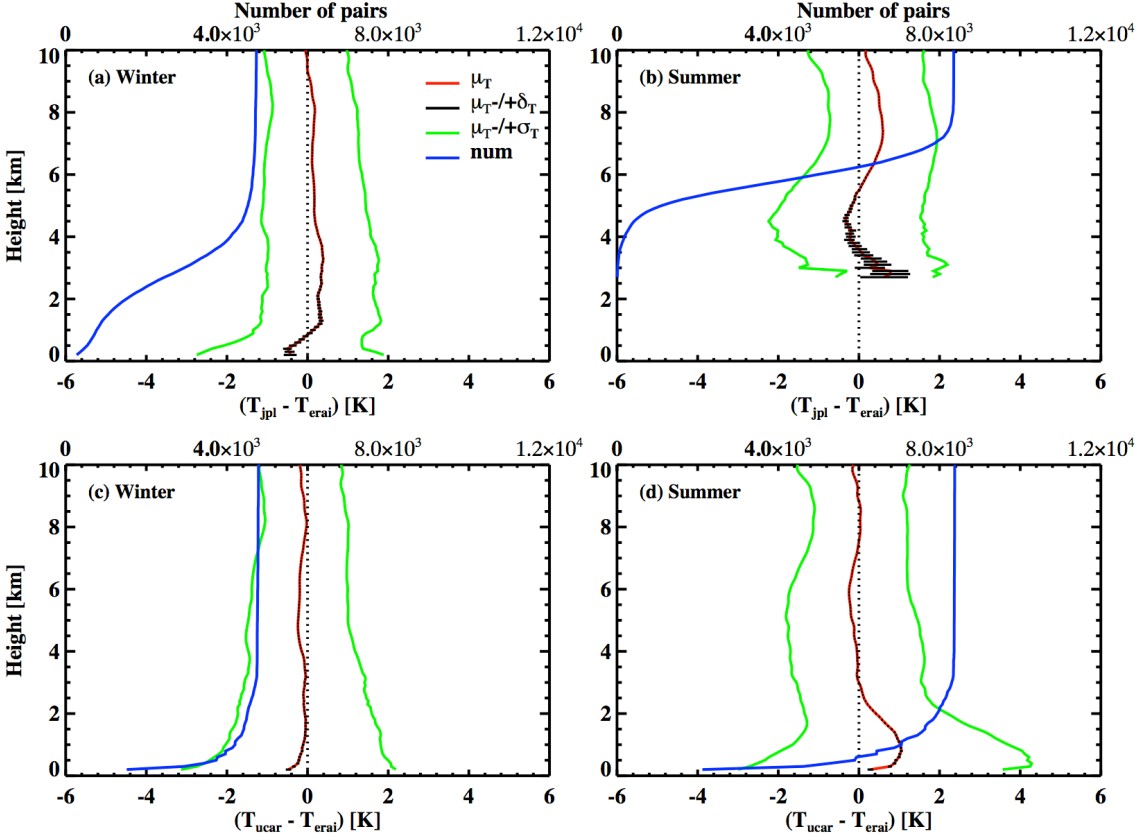

Figure 7: Same as Figure 6, except for temperature difference (K) between COSMIC RO and ERA-I.





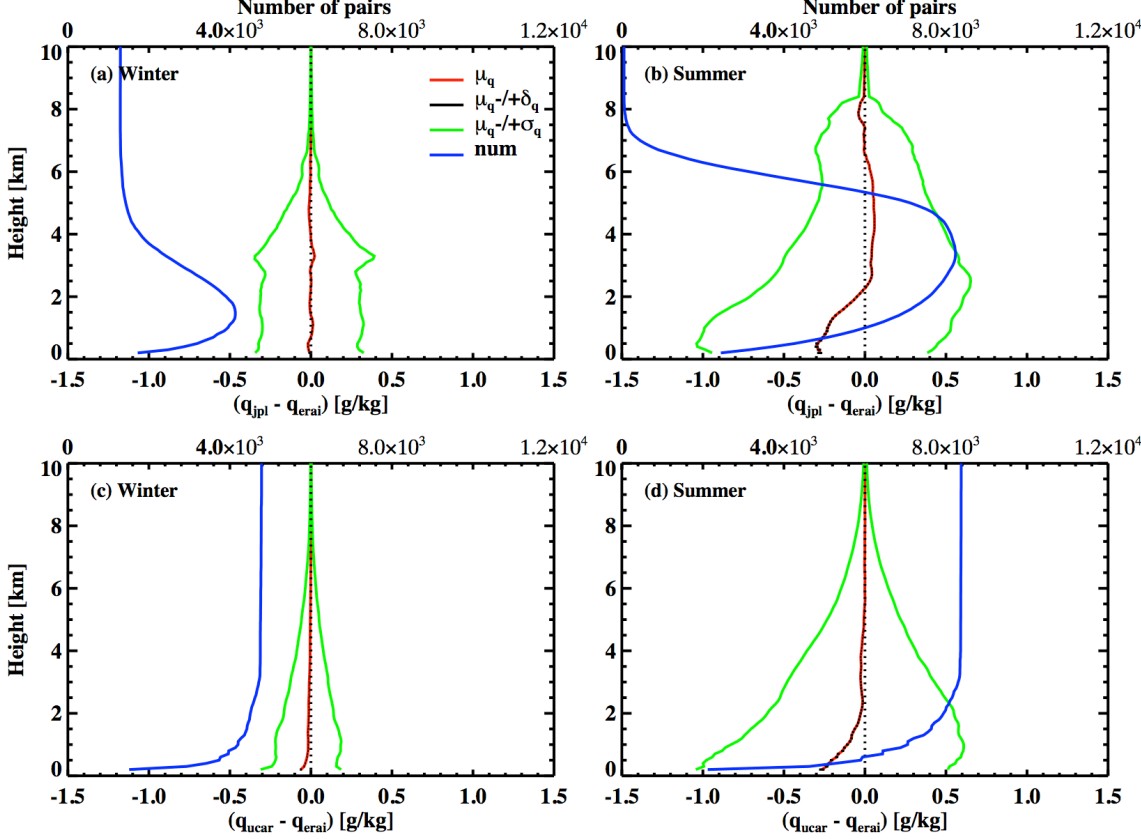

Figure 8: Same as Figure 6, except for specific humidity difference (g/kg) between COSMIC RO and ERA-I.

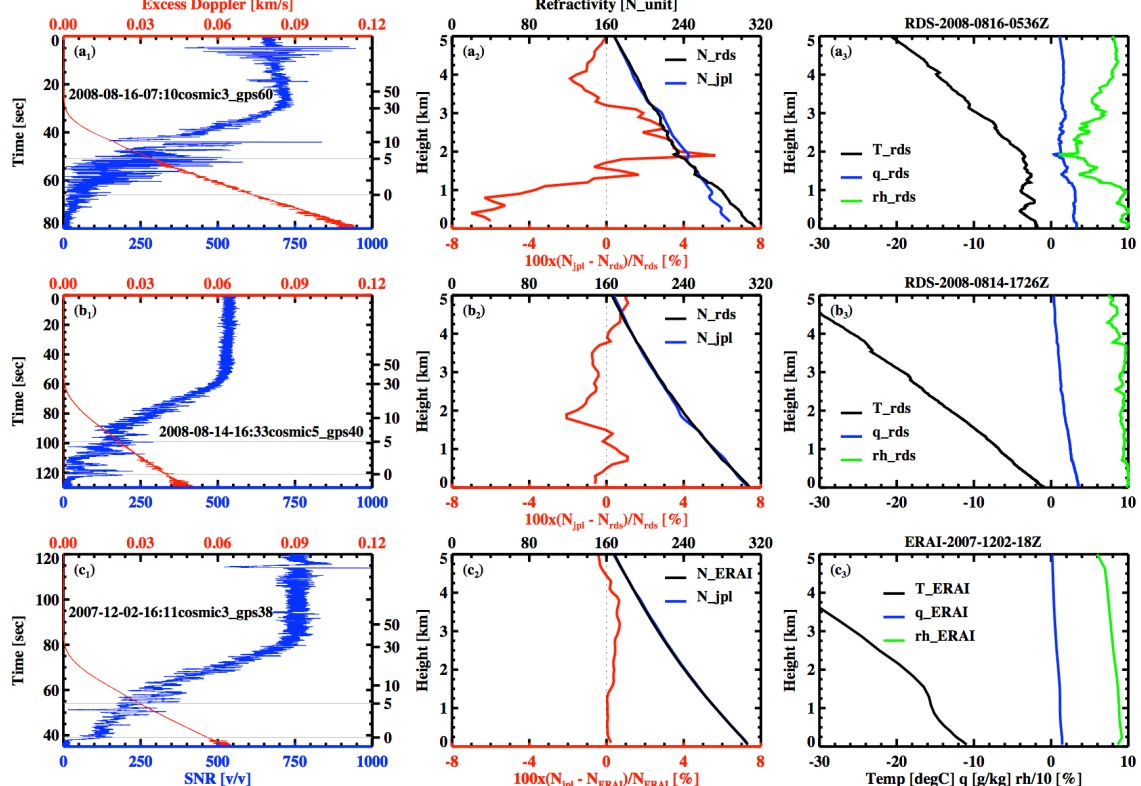

**Figure 9.** Three COSMIC RO soundings along with the near-coincident ASCOS radiosonde in summer (a, b), and ERA-I profile in winter (c). (Left) the COSMIC L1 SNR (blue) and excess Doppler (red) from JPL retrievals with the estimated tangent height shown on the right Y-axis, and the two horizontal thin lines indicating the surface and 5 km altitude; (Center) JPL RO refractivity (blue) and the near-coincident radiosonde and ERA-I profiles (black), along with the fractional refractivity difference (red); (Right) radiosonde and ERA-I profiles of temperature (black), specific humidity (blue) and the one-tenth of relative humidity (green).