# Peer review of "Evaluating the Lower Tropospheric COSMIC GPS Radio Occultation Sounding Quality over the Arctic"

_Atmospheric Measurement Techniques, 2017_

## Referee Comment (RC1) · Anonymous Referee #1 · 31 Oct 2017

General comments ——————-

I thank the authors for responding fully to my earlier points. The paper is now clearer and easier to read. The figures are generally clearer. I think it is now suitable for publication, subject to a few more clarifications.

Specific comments ———————-

Page 6, Line 14: The correct units of $a_1$ are N-units K / mb.

Page 6, Line 14: The correct units of $b_1$ are N-units K^2 / mb.

Page 7, Lines 14-16: I understand that the UCAR 1dvar retrieval depends on the errors in the RO measurement and the background field. But isn't the same thing true of the

[Figure]

JPL method? If I have understood Lines 1-5 of Page 7 correctly, this method amounts to using model fields to provide some of the terms in Eqn (1) so that the other fields can be deduced from it. In which case, doesn't the error on the derived T/q field depend on the error in the observation _and_ the error in the model q/T field too? If this is not so, please explain.

Technical corrections ————————-

Page 13, Line 15: COSIMC –> COSMIC

Page 13, Line 21: COMIC –> COSMIC

Page 31, Fig 9: I cannot see the horizontal lines at 0 and 5 km on my hardcopy. (They are clearer on the screen.) Please make them bolder or darker, and check that they are visible on hardcopy.

————————————————

---

## Referee Comment (RC2) · Anonymous Referee #3 · 31 Oct 2017

**General comments**

The paper "Evaluating the Lower Tropospheric COSMIC GPS Radio Occultation Sounding Quality over the Arctic" by Yu, Xie, and Ao investigates the usefulness of Radio Occultation (RO) observations for studies of the Arctic lower troposphere. The penetration depth of COSMIC RO sounding above 65 degrees north for different seasons is described, and the uncertainties of the lower-tropospheric RO observations are quantified using two type of uncertainty definitions: the structural uncertainty as quantified from the differences between two processing centers, and the parametric uncertainty as quantified by differences between RO and alternative data sources, here radio sondes and reanalysis data.

Overall, the results presented are mostly those expected. The impact of humidity

(mainly during Arctic summer) is line with the general understanding of RO behaviour. However, the study makes an important point in stressing the usefulness of RO for the lower troposphere in Arctic, where other data sources are scarce. RO data are most likely under-exploited in this type of application, and the manuscript gives a good description of the advantages and disadvantages of using RO data in the Arctic lower troposphere. The potential shortcomings of the RO method for this particular application are pointed out and at least partly explained.

The paper is suited for publication in AMT.

**Specific comment**
A question and a suggestion for the future: the parametric uncertainties of the JPL and UCAR data are obtained by identifying RO profiles closer than 3 hours and 300 km to a radiosonde profile. The statistics (mean, medians, and standard deviations) of the RO-radiosonde differences are plotted. These statistics are influenced both by actual RO-radiosonde differences and by the fact that the two type of observations are not perfectly collocated. We are mainly interested in the first component, while the second component acts to disguise the "true" differences. One could consider using the statistics of the double differences (RO-REANRO)-(RDS-REANRDS), where REANRO and REANRDS are reanalysis data collocated to the RO profiles and the radiosondes, respectively. The impact of an imperfect collocation between RO and radiosondes is much smaller with this type of comparison using a reanalysis model as a transfer medium. I would like to see a comment from the authors on this subject.

**Minor comments**
Abstract, line 10: "predication" should be "prediction".

Abstract, line 11: in "... demonstrated as a high-quality observation with ..." something appears to be missing. It should be "observation type" or "observation technique" rather than just "observation".

Abstract, line 12: "high-vertical-resolution" should be "high vertical resolution". It is correctly used in the Introduction, line 4, but not here. I'm not an expert in grammar, but this type of construction is only used as an adjective. As an example, this would be a correct sentence: "a measurement with a high vertical resolution is a high-vertical-resolution measurement."

Abstract, line 18: "Over 70

Introduction, line 1: "twice as much as global average rate" should be "twice as much as the global average rate".

"MetOp" should be "Metop".

---

## Author Comment (AC1) · 19 Jan 2018

Xiao et al., Evaluating the Lower Tropospheric COSMIC GPS Radio Occultation Sounding Quality over the Arctic

**We thank the editor and the two reviewers for the very insightful and constructive comments. A list of comments with detailed responses are shown below.**

**Response to the Anonymous Referee #1:**

General comments
1. I thank the authors for responding fully to my earlier points. The paper is now clearer and easier to read. The figures are generally clearer. I think it is now suitable for publication, subject to a few more clarifications.

**We thank the reviewer's very positive comments.**

Specific comments

1. Page 6, Line 14: The correct units of a1 are N-units K / mb.
   Page 6, Line 14: The correct units of b1 are N-units K^2 / mb.

   **Both equation 1 and 2 were updated to incorporate the values of the two constants into equations:**

   $$N = 77.6\frac{P}{T} + 3.73{\times}10^5\ \frac{P_w}{T^2}\ . \hspace{2cm} (1)$$

   $$N = 77.6\frac{P_{dry}}{T_{dry}} \hspace{3cm} (2)$$

2. Page 7, Lines 14-16: I understand that the UCAR 1dvar retrieval depends on the errors in the RO measurement and the background field. But isn't the same thing true of the JPL method? If I have understood Lines 1-5 of Page 7 correctly, this method amounts to using model fields to provide some of the terms in Eqn (1) so that the other fields can be deduced from it. In which case, doesn't the error on the derived T/q field depend on the error in the observation _and_ the error in the model q/T field too? If this is not so, please explain.

   **Yes, the reviewer is correct that the JPL retrievals are also affected by the errors from both the RO measurement and the a-priori model. However, JPL retrieval simply use a-priori temperature for moisture retrieval, or use a-priori moisture for temperature retrieval. The error characteristics in JPL retrieval could be slightly easier to interpret due to the simple method. On the other hand it is more challenging to separate the errors**

**resulting from RO measurement and a-priori model from the 1D-var retrieval used for UCAR retrieval.**

Technical corrections

1.  Page 13, Line 15: COSIMC –> COSMIC

    **Corrected.**

2.  Page 13, Line 21: COMIC –> COSMIC

    **Corrected.**

---

## Author Comment (AC2) · 19 Jan 2018

Xiao et al., Evaluating the Lower Tropospheric COSMIC GPS Radio Occultation Sounding Quality over the Arctic

**We thank the editor and the two reviewers for the very insightful and constructive comments. A list of comments with detailed responses are shown below.**

**Response to the Anonymous Referee #3:**

General comments
The paper "Evaluating the Lower Tropospheric COSMIC GPS Radio Occultation Sounding Quality over the Arctic" by Yu, Xie, and Ao investigates the usefulness of Radio Occultation (RO) observations for studies of the Arctic lower troposphere. The penetration depth of COSMIC RO sounding above 65 degrees north for different seasons is described, and the uncertainties of the lower-tropospheric RO observations are quantified using two type of uncertainty definitions: the structural uncertainty as quantified from the differences between two processing centers, and the parametric uncertainty as quantified by differences between RO and alternative data sources, here radio sondes and reanalysis data.

Overall, the results presented are mostly those expected. The impact of humidity (mainly during Arctic summer) is line with the general understanding of RO behaviour. However, the study makes an important point in stressing the usefulness of RO for the lower troposphere in Arctic, where other data sources are scarce. RO data are most likely under-exploited in this type of application, and the manuscript gives a good description of the advantages and disadvantages of using RO data in the Arctic lower troposphere. The potential shortcomings of the RO method for this particular application are pointed out and at least partly explained.

The paper is suited for publication in AMT.

**We thank the reviewer's very positive comments.**

Specific comments

1.  A question and a suggestion for the future: the parametric uncertainties of the JPL and UCAR data are obtained by identifying RO profiles closer than 3 hours and 300 km to a radiosonde profile. The statistics (mean, medians, and standard deviations) of the RO-radiosonde differences are plotted. These statistics are influenced both by actual RO-radiosonde differences and by the fact that the two type of observations are not perfectly collocated. We are mainly interested in the first component, while the second component acts to disguise the "true" differences. One could consider using the statistics of the double differences (RO-REANRO)-(RDS-REANRDS), where REANRO and REANRDS are reanalysis data collocated to the RO profiles and the radiosondes, respectively. The impact of an imperfect collocation between RO and

radiosondes is much smaller with this type of comparison using a reanalysis model as a transfer medium. I would like to see a comment from the authors on this subject.

**Following the reviewer's very interesting suggestion, we generated the statistics of the "double differencing" results between RO and radiosonde by using the collocated ERA-I reanalysis as the medium. Note the overall statistics are very similar to Fig. 5 in the manuscript. This further confirms the robustness of the error statistics in Fig. 5, which is not significantly affected by the spatial and temporal difference between the RO and the radiosonde dataset. However, a slight reduction of the errors below 2 km are seen in all three parameters (*N, T, q*), implying the slightly increases of collocation errors between the RO and radiosonde profiles in the lower troposphere, where larger spatial inhomogeneity in the atmosphere was expected.**

[Figure]

**Figure: Fractional difference between ASCOS radiosonde soundings and the near-coincident COSMIC RO from UCAR (top, $a_1$, $b_1$, $c_1$) and JPL (bottom, $a_2$, $b_2$, $c_2$), in terms of refractivity ($a_1$, $a_2$), temperature ($b_1$, $b_2$) and specific humidity ($c_1$, $c_2$). The median difference ($\mu$, red), the median difference plus-minus ($\pm$) median absolute deviation ($\sigma$, green) for all three parameters are also shown. The number of near-coincident pairs as a function of height is shown in blue with scale marked on the top to each panel. (Similar to Figure 5 in manuscript)**

Technical corrections

1. Abstract, line 10: "predication" should be "prediction".

   **Corrected.**

2. Abstract, line 11: in "... demonstrated as a high-quality observation with ..." something appears to be missing. It should be "observation type" or "observation technique" rather than just "observation".

**"technique" is added after "observation".**

3. Abstract, line 12: "high-vertical-resolution" should be "high vertical resolution". It is correctly used in the Introduction, line 4, but not here. I'm not an expert in grammar, but this type of construction is only used as an adjective. As an example, this would be a correct sentence: "a measurement with a high vertical resolution is a high-vertical-resolution measurement."

   **Corrected. Thanks.**

4. Abstract, line 18: "Over 70

   **The sentence is updated as follows:**
   **"Over 70% RO soundings penetrate deep into the lowest 300 m of the troposphere in all non-summer seasons. However, the fraction of such deep penetrating profiles reduces to only about 50-60% in summer, when near-surface moisture and its variation increase."**

5. Introduction, line 1: "twice as much as global average rate" should be "twice as much as the global average rate".

   **Corrected.**

6. "MetOp" should be "Metop".

   **Corrected.**

---

## Author Response (AR2)

**We thank the Associate Editor for the constructive comments. A list of comments with detailed responses are shown**
5 **below.**

**Response to the Associate Editor's Comments:**

Specific comments

1. page 6, line 19: you accidentally removed „250". Please correct to „T< 250 K".

   **Corrected.**

15 2. Table 2: The numbers for temperature differences T_JPL minus T_UCAR for 0–3 km in JJA2008 are somehow
   misleading as there are hardly any data below 3 km (compare Fig. 4d). I suggest to rather remove the numbers in the
   table. The same applies for specific humidity differences q_JPL minus q_UCAR above 5 km.

   **Corrected. The numbers were removed and the associated manuscript text was also updated. We also updated**
20 **the Table 2-4 with two decimal digits to be consistent with the text.**

3. Table 3: The same applies for numbers for specific humidity q_JPL minus q_RDS above 6 km.

   **Corrected. The numbers were removed and the associated manuscript text was also updated.**

[revised manuscript text omitted]